# Impact of the COVID-19 Pandemic on Emergency Department Workload and Emergency Care Workers’ Psychosocial Stress in the Outbreak Area

**DOI:** 10.3390/medicina57111274

**Published:** 2021-11-19

**Authors:** In-Hwan Yeo, Yun-Jeong Kim, Jong-Kun Kim, Dong-Eun Lee, Jae-Young Choe, Chang-Ho Kim, Jung-Bae Park, Kang-Suk Seo, Sin-Yul Park, Suk-Hee Lee, Jae-Kyung Cho, Sang-Hun Lee

**Affiliations:** 1Department of Emergency Medicine, School of Medicine, Kyungpook National University, Daegu 41944, Korea; inani1113@gmail.com (I.-H.Y.); kimyjem1@gmail.com (Y.-J.K.); delee@knu.ac.kr (D.-E.L.); choejy@hanmail.net (J.-Y.C.); 9754130@hanmail.net (C.-H.K.); jbpark@knu.ac.kr (J.-B.P.); kssuh@knu.ac.kr (K.-S.S.); 2Department of Emergency Medicine, College of Medicine, Yeungnam University, Daegu 42415, Korea; dryuri@naver.com; 3Department of Emergency Medicine, Daegu Catholic University, Daegu 42472, Korea; mycozzy@cu.ac.kr; 4Department of Emergency Medicine, Daegu Fatima Hospital, Daegu 41199, Korea; therealjk@hanmail.net; 5Department of Emergency Medicine, Keimyung University Dongsan Hospital, Daegu 42601, Korea; sanghun@dsmc.or.kr

**Keywords:** COVID-19, emergency department, healthcare workers, depression, anxiety, post-traumatic stress disorder

## Abstract

*Background and Objectives:* Due to the unexpected spread of coronavirus disease 2019 (COVID-19), there was a serious crisis of emergency medical system collapse. Healthcare workers working in the emergency department were faced with psychosocial stress and workload changes. *Materials and Methods*: This was a cross-sectional survey of healthcare workers in the emergency department in Daegu and Gyeongbuk, Korea, from November 16 to 25, 2020. In the survey, we assessed the general characteristics of the respondents; changes in the working conditions before and after the COVID-19 pandemic; and resulting post-traumatic stress disorder, depression and anxiety statuses using 49 questions. *Results*: A total of 529 responses were collected, and 520 responses were included for the final analyses. Changes in working conditions and other factors due to COVID-19 varied by emergency department level, region and disease group. Working hours, intensity, role changes, depression and anxiety scores were higher in the higher level emergency department. Isolation ward insufficiency and the risk of infection felt by healthcare workers tended to increase in the lower level emergency department. Treatment and transfer delay were higher in the fever and respiratory disease groups (M = 3.58, SD = 1.18; M = 4.08, SD = 0.95), respectively. In all the disease groups, both treatment and transfer were delayed more in Gyeongbuk than in Daegu. *Conclusions*: Different goals should be pursued by the levels and region of the emergency department to overcome the effects of the COVID-19 pandemic and promote optimal care.

## 1. Introduction

In November 2019, atypical pneumonia caused by the coronavirus disease 2019 (COVID-19) was reported in Wuhan, China. At first, it spread in Wuhan, China. However, shortly after, Daegu (metropolitan city) and Gyeongbuk (province adjacent to Daegu) in South Korea were some of the worst regions in the world concerning COVID-19 infection [1]. The first community-acquired infection in Korea occurred in Daegu on 18 February 2020. On 31 March, 81.6% (7984/9786) of the South Korean COVID-19 infections occurred in Daegu and Gyeongbuk, as the infection spread rapidly in religious groups and hospitals [2]. Since then, due to the unexpected COVID-19 pandemic in Korea and worldwide, healthcare workers (HCWs) working at the forefront have experienced changes in the treatment environment, such as work intensity, roles and hours. Medical resources such as personal protective equipment and isolation wards for treating infected patients have been found to be insufficient [3]. In addition, many emergency departments (EDs) have experienced temporary closures due to the COVID-19 infection [4].

In the world, studies on changes in stress, depression, anxiety and post-traumatic stress disorder (PTSD) caused by COVID-19 have been published [5,6,7]. However, studies on the changes in the treatment environment of EDs in Daegu and Gyeongbuk due to COVID-19 have not yet been published. Moreover, studies on the stress of HCWs in Korea are insufficient. As the pandemic continues, studies on the psychological stress of HCWs in Korea are now being published [8,9].

Accurate analysis of the main difficulties and stress factors faced by HCWs, who have experiences of directly working in the EDs is necessary to overcome the effects of the COVID-19 pandemic. Therefore, we studied what parts were difficult in the ED work due to the COVID-19 pandemic for HCWs in Daegu and Gyeongbuk who had experienced the pandemic initially in Korea. We also used anxiety, PTSD and depression scores to evaluate the HCWs’ psychological states. By studying the state of the medical environment and the psychological state of HCWs working at the forefront of the COVID-19 pandemic, a study was conducted to form the basis for establishing medical policies to overcome the COVID-19 pandemic.

## 2. Materials and Methods

This study was a cross-sectional survey conducted via text messages with a link to access Google surveys for physicians and nurses working in the EDs in Daegu and Gyeongbuk, Korea, from November 16 to 25, 2020. Responses arrived after the study period were excluded. We tried to contact all of 46 EDs in Daegu and Gyeongbuk [10]. However, three Level 3 EDs were temporarily or permanently closed. Finally, we contacted 43 EDs (5 Level 1 EDs, 10 Level 2 EDs and 28 Level 3 EDs, with Level 1 ED being the highest ED level) in Daegu and Gyeongbuk directly to confirm the number of HCWs and send them text messages. In the survey, sufficient information of the study was included on the first page, and only those who voluntarily agreed to the survey were included. We assessed the general characteristics of the respondents, changes in the working conditions before and after the COVID-19 pandemic and the PTSD, depression and anxiety statuses using 49 questions (Table 1).

To assess the changing working conditions, we used a 5-point Likert scale. We assessed PTSD using the Korean version of the Primary Care Post-Traumatic Stress Disorder Screen for the Diagnostic and Statistical Manual-5 (PC-PTSD-5; 0–1 = normal, 2 = mild, 3–5 = severe), depression using the Korean version of the Patient Health Questionnaire-9 (PHQ-9; 0–4 = normal, 5–9 = mild, 10–14 = moderate, 15–19 = moderately severe, 20–27 = severe) and anxiety using the Korean version of the Generalized Anxiety Disorder-7 Scale (GAD-7; 0–4 = normal, 5–9 = mild, 10–14 = moderate, 15–21 = severe) [11,12,13,14].

Data were analyzed using SPSS version 25 (IBM Corp., Armonk, NY, USA). In this study, the categorical variables are presented as numerals and percentages, whereas the continuous variables are presented as the mean and standard deviation (SD). Mann–Whitney U test and Kruskal–Wallis H test were used to compare the continuous variables. Statistical significance was set at *p* < 0.05. Bonferroni’s correction was applied to the post hoc analysis, and a Bonferroni-corrected *p*-value of *p* < 0.017 was used. Logistic regression analysis was performed to determine the risk factors for PTSD, depression and anxiety.

## 3. Results

The total number of willing participants among the HCWs (physicians and nurses) working in the ED in Daegu and Gyeongbuk was 1116. A total of 529 responses were collected, and nine incomplete or duplicate responses were excluded. Finally, 520 responses were included, resulting in a valid response rate of 46.6% (Table 1).

Questions were asked using a 5-point Likert scale to determine the changes in working conditions due to the COVID-19 pandemic. Changes in work intensity were the most highly scored change, with an average score of 3.85 (SD = 0.98). Treatment delay and transfer delay in patients with fever or respiratory symptoms were 3.58 (SD = 1.18) and 4.08 (SD = 0.95), respectively. The answer to “whether the isolation ward was sufficient” was 2.16 (SD = 0.98) on average, of which the lowest was 1.83 (SD = 0.96) in the Level 3 EDs. There were differences in work changes and isolation ward sufficiency according to the difference in the ED levels. There were statistically significant differences in treatment and transfer delays between Gyeongbuk and Daegu (Table 2).

A total of 225 (ED Work > Relationship > Personal) and 81 (ED Work > Personal > Relationship) respondents said that working in the ED was the most stressful thing concerning their personal life and interpersonal relationships when ED work after the COVID-19 pandemic had increased to 251 and 111, respectively (Table 3).

The mean depression score was 6.39 (SD = 5.60). Gender differences were found in the depression scores. The mean depression score for men was 5.46 (SD = 5.80), and the women’s mean depression score was 6.92 (SD = 5.42; *p* = 0.005). There was a difference in the depression scores according to differences in the ED levels. The mean depression scores of those working in Levels 1, 2 and 3 EDs were 7.81 (SD = 5.61), 6.67 (SD = 5.91) and 4.82 (SD = 4.76; *p* < 0.001), respectively. Anxiety scores also differed according to the ED level. The anxiety scores of Levels 1, 2 and 3 EDs were 4.31 (SD = 4.31), 4.11 (SD = 4.82) and 2.53 (SD = 3.45; *p* = 0.03; Figure 1, Table 4), respectively.

Logistic regression was used to examine the relationship between mild to severe PTSD (PC-PTSD-5 score ≥ 2), depression (PHQ-9 score ≥ 5) and anxiety (GAD-7 score ≥ 5) with gender, marital status, age, work experience, working region and ED level. Women reported higher depression scores (*p* = 0.007) than men. Those in higher-level EDs also reported higher depression (Level 1 vs. 2: *p* = 0.027, Level 1 vs 3: *p* < 0.001) and anxiety scores (Level 1 vs. 3: *p* = 0.005; Table 5) than those in lower-level EDs.

The results of whether there was a change in the working hours, work intensity, and role due to COVID-19 were 3.09 (SD = 1.07), 3.85 (SD = 0.98) and 3.32 (SD = 0.98), respectively, which showed a tendency to increase in the higher levels of the ED. The result of “risk of infection” was 3.51 (SD = 0.96), showing a tendency to increase in the lower levels of the ED.

## 4. Discussion

We conducted a study to overcome the COVID-19 pandemic efficiently by analyzing the workload and stress of HCWs in the EDs and to prepare for other pandemics that may occur in the future. Moreover, Daegu and Gyeongbuk are areas with high research value due to the early stage spread of the COVID-19 infection.

Our study showed that higher-level EDs suggested that HCWs were burdened with an increased workload, and lower-level EDs suggested an insufficient environment for treating infected patients, which could be interpreted as a situation where there was a risk of infection along with restrictions in treatment. Therefore, constructing more isolation rooms to accommodate infected patients and providing rapid COVID-19 PCR tools for a swift release from quarantine would be optimal solutions in the lower-level EDs. This will naturally decrease the concentration of patients and also reduce the work intensity of HCWs in the higher-level EDs. In addition, hiring more HCWs in the higher-level EDs would be one solution to decrease workload.

Comparing the treatment delay and transfer of patients in the five severe disease groups, HCWs complained of transfer delay more than treatment delay in all areas. Fever or respiratory disease was the only disease group with a treatment delay exceeding 3 with a score of 3.58 (SD = 1.18). Among the transfer delay scores, fever or respiratory disease showed the highest score of 4.08 (SD = 0.95) compared to the other severe disease groups. It could be suggested that insufficiency of isolation wards induced both treatment and transfer delay of fever or respiratory disease patients (Table 2). This suggests the need for sufficient isolation rooms for patients with fever or respiratory disease and rapid COVID-19 PCR tools for the swift release of patients from quarantine.

Gyeongbuk showed significant treatment and transfer delays compared to Daegu in all the severe disease groups. However, this study shows limitations in explaining this effect due to the lack of objective data. Therefore, an in-depth discussion with objective data is needed regarding the treatment and transfer issues of patients with fever and respiratory diseases and restrictions on treating and transferring patients with severe diseases in Gyeongbuk (Table 2). Additionally, if it is objectively proven, solutions to improve the treatment and transfer of patients in Gyeongbuk would be needed.

Studies related to the stress of HCWs due to the COVID-19 pandemic have been published [15]. In our study, the average value of PTSD and anxiety was within the normal range, but in the case of depression, the average value was 6.39 (SD = 5.6), which was in the mild depression (i.e., 5–9) range. The ratios of PTSD, depression and anxiety beyond the normal range were 206/520 (39.6%), 295/520 (56.7%) and 150/520 (28.8%), respectively.

There were no significant differences between occupations and regions, but there were differences in the depression scores according to gender (women > men). In addition, there were differences in the depression and anxiety scores according to the ED level (Level 1 > 2 > 3).

In general, depression appears at a higher rate in women than men due to biological factors and other complex reasons [16]. Moreover, many studies have shown that females have a higher depression score in studies conducted to measure depression caused by the COVID-19 pandemic [17,18,19]. Therefore, it is necessary to pay more attention to the gender differences in depression and anxiety in relation to the levels of ED, which were newly discovered in this study.

It is also noteworthy that HCWs in the higher-level EDs felt that the ED work was more stressful than their personal life and interpersonal relationships than the HCWs in the lower-level EDs. Therefore, it is necessary to pay attention to the working intensity and stress of HCWs in higher-level EDs, as well as discuss solutions that lead to an increasing number of HCWs and the lowering of patient concentration.

This study had several limitations. First, the list of HCWs working at the Daegu and Gyeongbuk EDs was directly investigated; thus, it may differ from the actual list. Second, there are possibilities of over-coverage, under-coverage and nonresponse errors. Third, this is a survey-based study, meaning there could be differences between the objective data and what the HCWs actually feel regarding work intensity, treatment delay and transfer delay. Finally, if a longitudinal study were to be conducted, there may be differences in the results depending on the period of the survey [20,21].

## 5. Conclusions

In conclusion, HCWs in the higher-level EDs were burdened with an increased workload and had increased depression and anxiety scores. HCWs in the lower-level EDs felt that isolation wards to treat infected patients were insufficient. HCWs at the Gyeongbuk ED experienced more delays in treatment and transfer than at Daegu. Moreover, treatment delay and transfer delay were higher in the fever and respiratory disease groups. Policymakers should be made aware of the differences in HCWs’ situations according to the ED levels and regions to overcome the effects of the COVID-19 pandemic, and further studies should be conducted to overcome the negative effects of the COVID-19 pandemic.

## Figures and Tables

**Figure 1 medicina-57-01274-f001:**
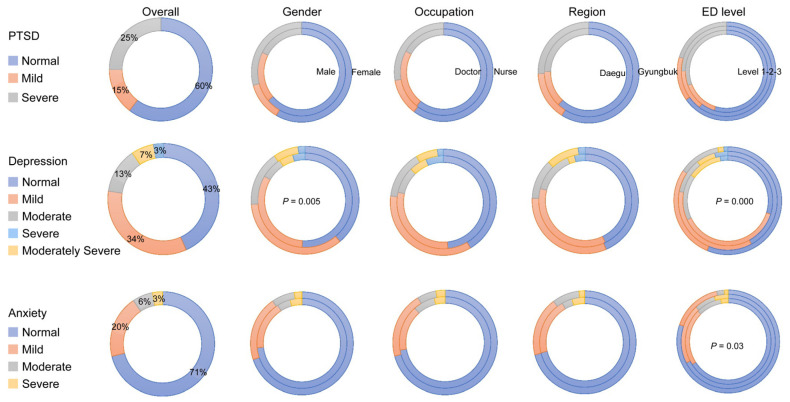
PTSD, depression and anxiety status of respondents. Values are presented as percentage (%). PTSD, post-traumatic stress disorder; ED, emergency department. PTSD: Korean version of the Primary Care Post-traumatic Stress Disorder Screen for the Diagnostic and Statistical Manual-5 (PC-PTSD-5), 0–1 = normal, 2= mild, 3–5 = severe. Depression: The Korean version of the Patient Health Questionnaire-9 (PHQ-9), 0–4 = normal, 5–9 = mild, 10–14 = moderate, 15–19 = moderately severe, 20–27 = severe. Anxiety: By the Korean version of generalized anxiety disorder-7 (GAD-7) scale, 0–4 = normal, 5–9 = mild, 10–14 = moderate, 15–21 = severe.

**Table 1 medicina-57-01274-t001:** Demographic characteristics of participants.

Variables	Total	Occupation	
		Nurse	Physician
Overall	520 (100)	390 (75.0)	130 (25.0)
Gender			
Male	188 (36.2)	71 (37.8)	117 (62.2)
Female	332 (63.8)	319 (96.1)	13 (3.9)
Marital status			
Married	202 (38.8)	115 (56.9)	87 (43.1)
Single	318 (61.2)	275 (75.0)	43 (25.0)
Age, yr			
All	33.51 ± 8.08	31.88 ±7.43	38.39 ± 7.98
Male	35.52 ± 8.03	29.93 ± 4.10	38.91 ± 7.94
Female	32.37 ± 7.89	32.31 ± 7.93	33.77 ± 7.11
Work experience, yr	8.20 ± 7.73	7.28 ± 7.30	10.95 ± 8.34
Working region			
Gyeongbuk	240 (46.2)	194 (80.8)	46 (19.2)
Daegu	280 (53.8)	196 (70.0)	84 (30.0)
ED			
Level 1	144 (27.7)	102 (70.8)	42 (29.2)
Level 2	209 (40.2)	149 (71.3)	60 (28.7)
Level 3	167 (32.1)	139 (83.2)	28 (16.8)

Values are presented as mean ± SD or number (%). SD = standard deviation; ED = emergency department.

**Table 2 medicina-57-01274-t002:** Changes in working conditions during the COVID-19 pandemic by ED and region.

	Total	ED			*p*	*p*-Value for Level 1 vs. 2	*p*-Value for Level 1 vs. 3	*p*-Value for Level 2 vs. 3	Region		*p*
		Level 1	Level 2	Level 3					Daegu	Gyeongbuk	
Work Change											
Increase in working hours	3.09 ± 1.07	3.40 ± 0.94	3.05 ± 1.08	2.87 ± 1.11	<0.001 *	0.001 **	<0.001 **	0.166	3.06 ± 1.10	3.12 ± 1.03	0.523
Increase in working intensity	3.85 ± 0.98	4.08 ± 0.78	3.94 ± 0.93	3.54 ± 1.11	<0.001 *	0.267	<0.001 **	0.008 **	3.80 ± 0.98	3.91 ± 0.98	0.177
Working role changes	3.32 ± 0.98	3.50 ± 0.85	3..33 ± 0.96	3.16 ± 1.08	0.026 *	0.076	0.006 **	0.137	3.27 ± 1.00	3.38 ± 0.94	0.178
Increased infection risk	3.51 ± 0.96	3.31 ± 0.95	3.52 ± 0.98	3.67 ± 0.90	0.008 *	0.645	0.408	0.171	3.52 ± 0.98	3.50 ± 0.93	0.832
Treatment Delay											
Acute coronary syndrome	2.47 ± 1.10	2.54 ± 1.14	2.54 ± 1.13	2.33 ± 1.02	0.164	0.908	0.102	0.095	2.32 ± 1.03	2.65 ± 1.16	0.001 *
Ischemic stroke	2.55 ± 1.12	2.76 ± 1.09	2.59 ± 1.18	2.31 ± 1.02	0.002 *	0.143	<0.001 **	0.028	2.37 ± 1.05	2.75 ± 1.17	<0.001 *
Hemorrhagic stroke	2.49 ± 1.11	2.66 ± 1.08	2.53 ± 1.16	2.28 ± 1.00	0.007 *	0.221	0.001 **	0.044	2.30 ± 1.00	2.70 ± 1.16	<0.001 *
Major trauma	2.61 ± 1.11	2.67 ± 1.12	2.70 ± 1.13	2.45 ± 1.07	0.068	0.790	0.079	0.028	2.44 ± 1.07	2.81 ± 1.13	<0.001 *
Fever or Respiratory disease	3.58 ± 1.18	3.42 ± 1.22	3.63 ± 1.18	3.66 ± 1.13	0.168	0.105	0.086	0.890	3.44 ± 1.21	3.75 ± 1.11	0.002 *
Transfer Delay											
Acute coronary syndrome	2.91 ± 1.14	2.76 ± 1.12	2.94 ± 1.14	3.01 ± 1.15	0.123	0.119	0.045	0.516	2.71 ± 1.13	3.14 ± 1.12	<0.001 *
Ischemic stroke	2.95 ± 1.15	2.83 ± 1.17	2.98 ± 1.16	3.02 ± 1.15	0.298	0.194	0.115	0.690	2.75 ± 1.13	3.20 ± 1.12	<0.001 *
Hemorrhagic stroke	2.97 ± 1.15	2.85 ± 1.13	3.00 ± 1.16	3.03 ± 1.15	0.357	0.206	0.141	0.751	2.77 ± 1.15	3.20 ± 1.11	<0.001 *
Major trauma	3.08 ± 1.15	2.90 ± 1.13	3.12 ± 1.16	3.19 ± 1.14	0.059	0.079	0.025	0.508	2.85 ± 1.16	3.35 ± 1.08	<0.001 *
Fever or respiratory disease	4.08 ± 0.95	3.87 ± 1.11	4.16 ± 0.87	4.17 ± 0.87	0.036 *	0.114	0.125	0.973	3.95 ± 1.04	4.23 ± 0.81	0.001 *
Facilities and Equipment											
Isolation ward	2.16 ± 0.98	2.47 ± 1.04	2.22 ± 0.96	1.83 ± 0.86	0.000 *	0.029	0.000 **	0.002 **	2.16 ± 0.99	2.17 ± 0.98	0.945
Personal protective equipment	3.21 ± 1.11	3.24 ± 1.09	3.31 ± 1.04	3.07 ± 1.19	0.139	0.705	0.187	0.053	3.31 ± 1.10	3.10 ± 1.11	0.031 *

ED, emergency department. Values are presented as mean ± SD (standard deviation). Likert scale: 1 = strongly disagree, 2 = disagree, 3 = neither agree nor disagree, 4 = agree, 5 = strongly agree. * *p* < 0.05. ** *p* < 0.017, Bonferroni corrected *p* value.

**Table 3 medicina-57-01274-t003:** Changes in the stress sequence before and after the pandemic.

	Before COVID-19 Pandemic				After COVID-19 Pandemic			
Stress Sequence	Level 1	Level 2)	Level 3	Total	Level 1	Level 2	Level 3	Total
ED Work > Relationship > Personal	74 (51.4)	87 (41.6)	64 (38.3)	225 (43.3)	80 (55.6)	97 (46.4)	74 (44.3)	251 (48.3)
ED Work > Personal > Relationship	26 (18.1)	29 (13.9)	26 (15.6)	81 (15.6)	35 (24.3)	45 (21.5)	31 (18.6)	111 (21.3)
Relationship > ED Work > Personal	25 (17.4)	45 (21.5)	34 (20.3)	104 (20.0)	12 (8.3)	33 (15.8)	36 (21.6)	81 (15.6)
Personal > ED Work > Relationship	9 (6.3)	29 (13.9)	16 (9.6)	54 (10.4)	9 (6.3)	22 (10.5)	17 (10.2)	48 (9.2)
Relationship > Personal > ED Work	6 (4.2)	11 (5.3)	11 (6.6)	28 (5.4)	5 (3.5)	5 (2.4)	2 (1.2)	12 (2.3)
Personal > Relationship > ED Work	4 (2.8)	8 (3.8)	16 (9.6)	28 (10.4)	3 (2.1)	7 (3.3)	7 (4.2)	17 (3.3)
*p*-Value	0.041				0.077			

ED, emergency department. Values are presented as number (%).

**Table 4 medicina-57-01274-t004:** PTSD, depression, and anxiety status of respondents.

	PTSD			Mean	Depression					Mean	Anxiety				Mean
	Normal	Mild	Severe		Normal	Mild	Moderate	Moderately Severe	Severe		Normal	Mild	Moderate	Severe	
Overall (*n* = 520)	314	76	130	1.47 ± 1.65	225	177	69	34	15	6.39 ± 5.60	370	101	33	16	3.66 ± 4.34
Gender															
Male	120	35	33	1.20 ± 1.51	96	61	14	9	8	5.46 ± 5.80	138	33	9	8	3.26 ± 4.58
Female	194	41	97	1.63 ± 1.71	129	116	55	25	7	6.92 ± 5.42	232	68	24	8	3.88 ± 4.19
*p*-Value	0.06				0.005 *						0.449				
Occupation															
Doctor	79	29	22	1.24 ± 1.56	63	39	12	8	8	6.22 ± 6.32	94	21	10	5	3.53 ± 4.79
Nurse	235	47	108	1.55 ± 1.67	162	138	57	26	7	6.45 ± 5.34	276	80	23	11	3.70 ± 4.19
*p*-Value	0.362				0.371						0.903				
Region															
Gyeongbuk	149	32	59	1.39 ± 1.61	103	87	35	6	9	6.35 ± 5.62	171	49	9	11	3.73±4.56
Daegu	165	44	71	1.54 ± 1.68	122	90	34	28	6	6.43 ± 5.59	199	52	24	5	3.59 ± 4.15
*p*-Value	0.547				0.653						0.917				
ED															
Level 1	79	21	44	1.71 ± 1.69	43	55	24	18	4	7.81 ± 5.61	95	31	14	4	4.31 ± 4.31
Level 2	126	31	52	1.49 ± 1.67	88	75	24	13	9	6.67 ± 5.91	141	43	15	10	4.11 ± 4.82
Level 3	109	24	34	1.25 ± 1.57	94	47	21	3	2	4.82 ± 4.76	134	27	4	2	2.53 ± 3.45
*p*-Value	0.118				0.000 *						0.03 *				

Values are presented as mean ± SD or number (%). SD, standard deviation; PTSD, post-traumatic stress disorder; ED, emergency department. PTSD rated by Korean version of the Primary Care Post-Traumatic Stress Disorder Screen for the Diagnostic and Statistical Manual-5 (PC-PTSD-5), 0–1 = normal, 2 = mild, 3–5 = severe. Depression rated using the Korean version of the Patient Health Questionnaire-9 (PHQ-9), 0–4 = normal, 5–9 = mild, 10–14 = moderate, 15–19 = moderately severe, 20–27 = severe. Anxiety rate by the Korean version of Generalized Anxiety Disorder-7 Scale (GAD-7), 0–4 = normal, 5–9 = mild, 10–14 = moderate, 15–21 = severe. * *p* < 0.05.

**Table 5 medicina-57-01274-t005:** Factors associated with depression and anxiety.

	Variables	B	Standard Error	Wald	*p*	aOR	95% CI
Depression *	Gender (Male = 0)						
	Male vs. Female	0.514	0.189	7.406	0.007	1.672	1.155–2.421
	ED Level (Level 1 = 0)						
	Level 1 vs. Level 2	−0.512	0.231	4.896	0.027	0.599	0.381–0.943
	Level 1 vs. Level 3	−1.115	0.242	21.319	0	0.328	0.204–0.526
Anxiety **	ED Level (Level 1 = 0)						
	Level 1 vs. Level 2	−0.067	0.23	0.086	0.77	0.935	0.596–1.467
	Level 1 vs. Level 3	−0.739	0.262	7.955	0.005	0.477	0.286–0.798

To determine the logistic model calibration, we calculated the Hosmer-Lemeshow goodness of fit (*p* value of depression = 0.505, *p* value of anxiety = 1). aOR, adjusted odds ratio; ED, emergency department. * Depression: Korean version of the Patient Health Questionnaire-9 (PHQ-9), 0–4 = normal, 5–9 = mild, 10–14 = moderate, 15–19 = moderately severe, 20–27 = severe. ** Anxiety: By the Korean version of generalized anxiety disorder-7 (GAD-7), 0–4 = normal, 5–9 = mild, 10–14 = moderate, 15–21 = severe.

## Data Availability

Not applicable.

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
