# Peer review of "Impact of the COVID-19 Pandemic on Emergency Department Workload and Emergency Care Workers’ Psychosocial Stress in the Outbreak Area"

_medicina, 2021, doi:10.3390/medicina57111274_

Round 1

Reviewer 1 Report

TITLE: I am not sure that is important to have names of the cities in the title. Title present your work, it is usual to have study design.

Abstract: do not use abbreviations in the abstract

Introduction: what is international scientific contribution of your paper. You stated local contribution but this journal aim to have wider public, and you need to better describe this point.

Materials and methods: Sample should be better described, how do you define sample?  Ethical issues are not clear, is there informed consent it was voluntarily or not. Even you have at the end of manuscript approval from your board, you have to explain better in this section. Is use of instruments approved form authors? Have you performed psychometric validation of tests?

Results are presented clear. 

Discussion - you should start with your study aim and present the major findings. 

Conclusion - well

Author Response

Response to Reviewer 1 Comments

We appreciate the thoughtful and positive reviews.

In this reply you will find a point-by-point response to each of the reviewer’s concerns.

We hope that the revised manuscript will now be suitable for publication, and thank you to the reviewer.

Point 1

TITLE: I am not sure that is important to have names of the cities in the title. Title present your work, it is usual to have study design.

Response 1:

Thank you for your comment. Based on your comment, we changed the title without the city names. Our revised title is “Impact of the COVID-19 Pandemic on Emergency Department Workload and Emergency Care Workers’ Psychosocial stress in a local area”.

Point 2

Abstract: do not use abbreviations in the abstract

Response 2:

Thank you for your comment. We changed the all abbreviations to full terms in the abstract except COVID-19.

Point 3

Introduction: what is international scientific contribution of your paper. You stated local contribution but this journal aim to have wider public, and you need to better describe this point.

Response 3:

Thank you for your comment.

In the early days of the COVID-19 pandemic, Daegu was called ‘second Wuhan’, because it was one of the most prevalent area after Wuhan. At that time, there was not enough information about COVID-19.  There was also a lack of facilities and equipment to treat COVID-19 patients. Because of this, HCWs in the Daegu and Gyeongbuk suffered from extreme stress and heavy workload. To study the effect of COVID-19 pandemic on HCW’s stress and workload, Daegu and Gyeongbuk are important regions in the world.

So following sentences and reference were added and reviesed in the introduction.

“In November 2019, atypical pneumonia caused by the coronavirus disease 2019 (COVID-19) was reported in Wuhan, China. At first, it has spread out in Wuhan, China. But, shortly after then, Daegu (metropolitan city) and Gyeongbuk (province adjacent to Daegu) in the South Korea were one of the worst region in the world that COVID-19 infection was spread out [1]. The first community-acquired infection in the Korea occurred in Daegu on February 18, 2020. On March 31, 81.6% (7984/9786) of the South Korean COVID-19 infections occurred in Daegu and Gyeongbuk, as the infection spread rapidly in religious groups and hospitals [2].”

And, following sentences were added in the discussion.

“Also Daegu and Gyeongbuk are areas with high research value due to the early stages spread of the COVID-19 infection.”

Point 4

Materials and methods: Sample should be better described, how do you define sample? 

Response 4:

Thank you for your comment. We described more clearly about study sample. The following sentences were added and revised.

“We tried to contact all of 46 EDs in Daegu and Gyeongbuk [10]. But, three Level 3 EDs were temporarily or permanently closed. Finally, we contacted 43 EDs (5 Level 1 EDs, 10 Level 2 EDs, and 28 Level 3 EDs, with Level 1 ED being the highest ED level) in Daegu and Gyeongbuk directly to confirm the number of HCWs and send them text messages.”

Point 5

Ethical issues are not clear, is there informed consent it was voluntarily or not. Even you have at the end of manuscript approval from your board, you have to explain better in this section. Is use of instruments approved form authors? Have you performed psychometric validation of tests?

Response 5:

Thank you for your comment.  We conduced survey by the Google survey. In the survey, we provided sufficient explanations, and we only got responses with the voluntary participation of those who agreed to it. So following sentences were added.

“In the survey, sufficient information of the study was included on the first page, and only those who voluntarily agreed to the survey were conducted”

Also, we used validated psychometric tool. References were included at number 11 to 14.

Point 6

Results are presented clear.

Response 6:

Thanks for the good review.

Point 7

Discussion - you should start with your study aim and present the major findings.

Response 7:

Thank you for your comment. Based on your comment, the following sentences were added.

“We conducted a study to overcome the COVID 19 pandemic efficiently by analyzing the workload and stress of HCWs in the EDs and to prepare for other infectious era that may occur in the future. Also, Daegu and Gyeongbuk are areas with high research value due to the early stages spread of the COVID-19 infection.”

Point 8

Conclusion – well

Response 8:

Thanks for the good review.

Reviewer 2 Report

Firstly, thank you for opportunity to review very interested article. I don't feel qualified to judge about the English language and style

  1. The title reflect the main subject about workload and stress in ED during COVID-19, title was clear and easy to understand
  2. The abstract summarize and reflect the work described in the manuscript
  3. The key words reflect the focus of the manuscript. However I suggested to use full term of ED and PTSD.
  4. The manuscript adequately describe the background, present status, and significance of the study. The authors explain COVID-19 pandemic in Korea. I suggest the authors explain about stress or workload in other country to identify the problems of study.
  5. In line 50 "Eds" was incorrected, please revised.
  6. The manuscript describe methods in adequate detail, study subjects were clear, with demonstrate IRB number or text to human ethics consideration. 
  7. In Materials and Methods, I suggested the authors to explain about "text messages" in detail, the time for wait to response after sending message, if not response within time the authors re-sending? This study had inclusion or exclusion criteria? HCWs in ED include only doctor or nurse? 
  8. The manuscript interpret the findings adequately and appropriately, highlighting the key points concisely, clearly, and logically.
  9. Tables and figures sufficient, good quality and appropriately illustrative of the paper contents.
  10. The manuscript meet the requirements of biostatistics.
  11. The manuscript cite appropriately the latest, important and authoritative references in the introduction and discussion sections.

Author Response

Response to Reviewer 2 Comments

We appreciate the thoughtful and positive reviews.

In this reply you will find a point-by-point response to each of the reviewer’s concerns.

We hope that the revised manuscript will now be suitable for publication, and thank you to the reviewer.

Point 1

Firstly, thank you for opportunity to review very interested article. I don't feel qualified to judge about the English language and style

Response 1:

Thanks you. Our study was modified by professional English correction center.

Point 2

The title reflect the main subject about workload and stress in ED during COVID-19, title was clear and easy to understand

Response 2:

Thanks for the good review.

Point 3

The abstract summarize and reflect the work described in the manuscript

Response 3:

Thanks for the good review.

Point 4

The key words reflect the focus of the manuscript. However I suggested to use full term of ED and PTSD.

Response 4:

Thank you for your comment. Based on your comment, we changed the ED and PTSD to full terms in the Keywords. ED changed to Emergency Department. PTSD changed to post-traumatic stress disorder.

Point 5

The manuscript adequately describe the background, present status, and significance of the study. The authors explain COVID-19 pandemic in Korea. I suggest the authors explain about stress or workload in other country to identify the problems of study.

Response 5:

Thank you for your comment. Based on your comment, we explained more about in other country.

 So following sentences and reference were added.

“In the world, studies on changes in stress, depression, anxiety and PTSD caused by COVID-19 have been published [5-7].”

Point 6

In line 50 "Eds" was incorrected, please revised.

Response 6:

Thank you for your comment. We revised “Eds” to “EDs”. 

Point 7

The manuscript descripe methods in adequate detail, study subjects were clear, with demonstrate IRB number or text to human ethics consideration.

Response 7:

Thanks for the good review.

Point 8

In Materials and Methods, I suggested the authors to explain about "text messages" in detail, the time for wait to response after sending message, if not response within time the authors re-sending? This study had inclusion or exclusion criteria? HCWs in ED include only doctor or nurse?

Response 8:

Thank you for your comment. We explained more about text messages. We send messages just one time to physicians and nursed working in the ED. And responses arrived after the study period were excluded.

So following sentences and reference were added.

“This study was a cross-sectional survey conducted via text messages with a link to access Google surveys for physicians and nurses working in the EDs in Daegu and Gyeongbuk, Korea, from November 16 to 25, 2020. Responses arrived after the study period were excluded.”

Point 9

The manuscript interpret the findings adequately and appropriately, highlighting the key points concisely, clearly, and logically.

Response 9:

Thanks for the good review.

Point 10

Tables and figures sufficient, good quality and appropriately illustrative of the paper contents.

Response 10:

Thanks for the good review.

Point 11

The manuscript meet the requirements of biostatistics.

Response 11:

Thanks for the good review.

Point 12

The manuscript cite appropriately the latest, important and authoritative references in the introduction and discussion sections.

Response 12:

Thanks for the good review.

Round 2

Reviewer 2 Report

Thank you for revised version.